# OpenReview forum: "AdvWave: Stealthy Adversarial Jailbreak Attack against Large Audio-Language Models"
_ICLR.cc/2025/Conference — ICLR 2025 Poster_

### Official Review · Reviewer_cSgt · 2024-10-20

**Soundness:** 3
**Presentation:** 3
**Contribution:** 2
**Rating:** 3
**Confidence:** 5

**Summary:**

This paper presents an innovative adversarial attack method targeting LALMs, marking the first successful attack on LALMs with optimized audio stealth. The efforts are commendable.

**Strengths:**

The work is groundbreaking as it introduces the first jailbreaking adversarial attack on LALMs. The authors have conducted extensive experimental comparisons, particularly by adapting jailbreaking methods from other domains to the audio sector to ensure the superiority of the proposed attack method. The contribution of this paper is indisputable. However, I still have some questions regarding the audio stealthiness mentioned by the authors.

**Weaknesses:**

I believe the design motivation behind the authors' idea might be flawed. The audio provided clearly contains malicious content, so why consider the stealthiness of the adversarial disturbance? A normal listener would already notice something amiss with the content. Adding adversarial noise to silence segments inevitably leads to listeners hearing malicious content followed by eerie noises, which is utterly unconvincing from a realistic perspective. The authors should more reasonably consider the reasons for the stealthiness of adversarial disturbances and integrate them with the application scenarios of LALMs for a rational design.

**Questions:**

In the supplementary materials provided, I am puzzled about adding adversarial noise: 1. The authors mention that the adversarial noise is naturalized using urban environmental sounds as a masking method. However, I can still hear the traditional adversarial disturbances beyond the environmental sounds, suggesting the presence of two types of perturbations, which the paper does not mention. 2. The attack audio samples provided have adversarial disturbances implanted at the end silence segments of the audio, occupying about half the duration of the audio itself. It's unlikely for such a high proportion of silence in most audio datasets, revealing a serious issue: can adversarial attacks unrestrictively zero-pad benign audio ensure attack success? This seems to relate to the authors' initial claim that audio attacks on LALMs would limit the optimization search space for adversarial disturbances. I imagine the authors extended the audio to ensure sufficient search space, yet this seems impractical in real situations. 3. I am curious why the adversarial disturbances were added to the silence segments. Semantically rich portions of the audio seem more susceptible to attacks, and placing disturbances in silent parts would make the noise more detectable by human ears.

---

> ### Author Response · Authors · 2024-11-25
> **Response to Reviewer cSgt**
>
> We appreciate the reviewer's thoughtful feedback on our paper. Below, we included additional comments to further improve our work.
>
> > Q1: More discussions of why we want to enforce stealthiness in jailbreaks.
>
> Thank you for the suggestion! The objective of enforcing stealthiness during optimization is motivated by empirical observations. Without the stealthiness constraint, the optimized adversarial audio, while effective, often sounds screechy. This unnatural quality draws undue attention from human auditors and risks being flagged or filtered by noise-detection systems. For illustration, we include examples of adversarial audio without the stealthiness constraint in the supplementary material.
> By enforcing stealthiness, we aim to make the adversarial audio sound natural, minimizing suspicion and avoiding detection by noise filters. This motivation aligns with text-based jailbreaks, where recent works [1,2] enhance fluency and readability of adversarial prompts to bypass perplexity-based filters.
> We included the discussions into Section 3.1.
>
> [1] Guo, Xingang, et al. "Cold-attack: Jailbreaking llms with stealthiness and controllability." ICML 2024.
>
> [2] Sadasivan, Vinu Sankar, et al. "Fast Adversarial Attacks on Language Models In One GPU Minute." ICML 2024.
>
> > Q2: Why do we add the adversarial segment as a suffix instead of adding it to the original query? The length of the adversarial suffix is non-trivial. The audio sample provided also contains white noises in addition to natural sounds.
>
> Thank you for the insightful question!
> We would like to clarify that, similar to the approach in jailbreak literature where adversarial suffixes are added [3] instead of modifying the original tokens, our reasoning is as follows: altering the original query could lead to semantic changes, which might result in false positive jailbreaks. For example, if the query “How to use a gun for fun?” is modified to “How to use a water gun for fun?”, the model is likely to respond concretely, but this would not qualify as a successful jailbreak. Thus, in the context of LALM jailbreaks, we aim to keep the original query fixed to preserve its semantics and avoid such false positives.
>
> In response to your comments on audio suffix lengths, we have included ablation studies of adversarial suffixes with varying lengths in Table A. The results indicate that when the adversarial suffix length exceeds 50 audio frames, AdvWave becomes less sensitive to further changes in suffix length while consistently demonstrating high ASR (attack success rates). For context, one audio token typically corresponds to approximately 0.05 seconds, assuming standard sampling rates and window sizes. Consequently, the audio suffix lengths we tested are within reasonable ranges. Since the suffix is masked by natural sounds, it typically resembles background noise, enhancing its stealthiness.
>
> Lastly, thank you for reviewing our audio samples and highlighting the presence of white noise alongside the natural sounds in the optimized audio. We believe this noise is within acceptable limits and does not significantly impact stealthiness, as evidenced by our quantified stealthiness score. Nonetheless, incorporating additional smoothness penalties into our optimization framework could potentially address this issue. We have clarified this point in Section 3.4.
>
>
> Table A: Attack success rates ASR-W and ASR-L with SpeechGPT on Advbench-Audio dataset.
>
> | Length of adversarial audio suffix  | 10     | 30     | 50     | 70     | 90     |
> |----------|--------|--------|--------|--------|--------|
> | ASR-W    | 0.296  | 0.499  | 0.643  | 0.676  | 0.699  |
> | ASR-L    | 0.245  | 0.563  | 0.603  | 0.621  | 0.633  |
>
>
> [3] Zou, Andy, et al. "Universal and transferable adversarial attacks on aligned language models." arXiv preprint arXiv:2307.15043 (2023).

---

> > ### Comment · Reviewer_cSgt · 2024-11-25
> > **Response to author**
> >
> > While the authors argue that existing jailbreak attacks are predominantly achieved by appending suffixes, this is not entirely accurate [1]. Incorporating adversarial perturbations directly into the main body of the audio—rather than appending extra audio suffixes—is a widely adopted approach in current audio attacks [2-4]. This method allows attackers to modify the semantic content of the audio as a whole, thereby avoiding the need to add additional audio segments.
> >
> > This approach also facilitates the realization of low-perturbation audio attacks, which are less likely to be detected by human auditors due to the subtle nature of the perturbations. In contrast, generating audio with high perturbation levels, such as noticeable appended suffixes, increases the likelihood of detection.
> >
> > Given these points, the author’s response is unconvincing as it does not address these practical and commonly utilized techniques in audio adversarial attacks.
> > [1] Yang, Yijun, et al. "Mma-diffusion: Multimodal attack on diffusion models." Proceedings of the IEEE/CVF Conference on Computer Vision and Pattern Recognition. 2024.
> > [2] Qin, Yao, et al. "Imperceptible, robust, and targeted adversarial examples for automatic speech recognition." International conference on machine learning. PMLR, 2019.
> > [3] Yakura H, Sakuma J. Robust audio adversarial example for a physical attack[J]. arXiv preprint arXiv:1810.11793, 2018.
> > [4] Carlini N, Wagner D. Audio adversarial examples: Targeted attacks on speech-to-text[C]//2018 IEEE security and privacy workshops (SPW). IEEE, 2018: 1-7.

---

> > > ### Author Response · Authors · 2024-11-26
> > > **Follow-up discussion with Reviewer cSgt**
> > >
> > > We appreciate the reviewer’s feedback, which has prompted us to reconsider our threat model—specifically, the approach of adding perturbations or appending a suffix. We acknowledge that the references provided focus on adversarial attacks on ALMs rather than jailbreaks. However, jailbreak literature on LLMs (e.g., [1,2,3,4]) predominantly adopts the strategy of appending a suffix to the original query while preserving its semantics.
> > >
> > > This preference stems from the need to maintain the query’s original meaning, thereby avoiding false positive jailbreaks. For instance, in the example provided in our rebuttal, altering “gun” to “water gun” detoxifies the query and results in a false positive jailbreak. Consequently, we maintain that appending a jailbreak suffix is a more appropriate and reliable threat model for jailbreaking ALMs.
> > >
> > >
> > > [1] Guo, Xingang, et al. "Cold-attack: Jailbreaking llms with stealthiness and controllability." ICML 2024.
> > >
> > > [2] Sadasivan, Vinu Sankar, et al. "Fast Adversarial Attacks on Language Models In One GPU Minute." ICML 2024.
> > >
> > > [3] Qin, Yao, et al. "Imperceptible, robust, and targeted adversarial examples for automatic speech recognition." International conference on machine learning. PMLR, 2019.
> > >
> > > [4] Zou, Andy, et al. "Universal and transferable adversarial attacks on aligned language models." arXiv preprint arXiv:2307.15043 (2023).

---

> > > ### Author Response · Authors · 2024-12-03
> > >
> > > Dear Reviewer cSgt,
> > >
> > > As the discussion period draws to a close, we would like to kindly inquire if you have any further questions or feedback regarding our work. Your insights and suggestions have been valuable in helping us improve our work.
> > >
> > > Additionally, we hope you might consider reevaluating your rating in light of the updates and clarifications provided during the rebuttal process.
> > >
> > > Thank you again for your time and thoughtful review.
> > >
> > > Sincerely,
> > > Authors of AdvWave

---

### Official Review · Reviewer_7mqQ · 2024-11-01

**Soundness:** 1
**Presentation:** 2
**Contribution:** 3
**Rating:** 3
**Confidence:** 4

**Summary:**

This paper introduces AdvWave, a framework for conducting white-box adversarial attacks against large audio-language models (LALMs) to elicit harmful information. The authors identify the unique challenges posed by LALMs, such as gradient shattering due to discretisation operations in audio encoders and maintaining stealthiness constraints. To address these issues, AdvWave implements a dual-phase optimisation strategy. The first phase optimises a discrete latent representation to circumvent the gradient shattering issue, while the second phase adjusts the audio waveform itself to align closely with this representation while preserving perceptual naturalness. AdvWave significantly outperforms transferring static jailbreak attacks optimised on text-only LLMs that are subsequently vocalised with text-to-speech (TTS).  The authors argue that their approach highlights the need for more robust defenses and safety measures in LALMs.

**Strengths:**

- **Relevant and timely approach**: LALMs are becoming more prevalent with the recent release of audio capabilities in Gemini and GPT-4o. However, to the best of my knowledge, AdvWave is the first work that has successfully got white-box jailbreak attacks to work.
- **Innovative approach**: AdvWave uses a dual-phase optimisation strategy to address the issue of not being able to backpropagate through the full network when it contains discretisation. They also improve optimisation efficiency by adaptively finding a target string that matches the common structure the model uses for benign requests. These challenges are clearly explained, and the authors provide solutions.
- **Potential for future research**: AdvWave opens several avenues for future work, including further exploration of defensive mechanisms and applying the framework to LALMs that do not have a discretization bottleneck.

**Weaknesses:**

- **AdvWave is not the first approach to jailbreaking LALMs:** The authors of the paper claim that AdvWave is a novel framework for jailbreaking LALMs, but I think this claim needs to be softened throughout the paper. Gemini and GPT4o both have audio-specific jailbreaks, where they vocalise text jailbreaks with TTS in their recent model cards. Therefore, claiming that AdvWave is a novel white-box attack methodology for LALMs is better.
- **Stealthiness constraints are not well-motivated or measured:** It isn’t clear to me why this stealthiness constraint is needed. Any jailbreak, no matter the stealthiness of the input, is bad. Also, stealthiness constraints are not new; they were used in white-box attacks of speech-to-text (STT) models, and the intro doesn’t explain why LALMs make it more difficult. Papers such as “Imperceptible, robust, and targeted adversarial examples for automatic speech recognition” should be cited. Did you ablate changing the whole audio file rather than just the suffix? What motivated the suffix and the environmental noises? You can have imperceptible changes to the audio without that, I believe. Also, what environmental classifier do you use? This needs to be cited for replication purposes.
    - I’m very confused by the stealth metric and why it is a useful comparison to the baselines. The baselines do not have adversarial suffixes added on; they are just TTS reading a harmful prompt. So why isn’t their S_stealth equal to 1? It should be maximally stealthy, just like the vanilla baseline. Also, the baselines do not have a car horn at the end of the utterance, which could be considered more stealthy than your method. You mention you need high stealthiness so it is less detectable by real-world guardrail systems, but I don’t think the results presented demonstrate this. Also, AdvWave is not superior to vanilla in terms of stealth. The three terms in S_stealth are confusing and not well-motivated.
- **Lack of relevant adaptive black-box baselines:** The paper only compares attacks that are optimised on text-only LLMs that are then transferred into audio with TTS. Using TTS to vocalise GCG attacks might not make sense - there could be lots of tokens that can’t be spoken properly so I would expect the attack to be very weak. You say there are no adaptive attacks due to gradient shattering, but there are plenty of good adaptive black box attacks. I expect PAIR and TAP to work well in the audio domain. AdvWave should be evaluated against much stronger baselines than currently used. How strong are the transfer GCG/BEAST attacks to the base text-only LLM? E.g. what is the GCG ASR on Llama3? That would inform if the baselines transfer to the audio domain effectively or if they are broken by the architectural / fine-tuning differences.
- **Lack of clarity on LALM architecture differences, what architecture AdvWave is aimed at, and motivation for why dual optimisation to solve gradient shattering is needed:** Not all LALMs have discretisation of audio before input to an LLM (like SpeechGPT). Many insert a continuous vector directly into the embedding space of the model (e.g. DiVA, Llama3.1, Salmonn, Llasm, AudioQwen). Therefore, these won’t have the gradient shattering problem, and the framework in Figure 1 isn’t relevant.  There needs to be better motivation and explanation of why AdvWave targets LALMs that have the discrete bottleneck. Ideally, the paper will explain all the different architectures and introduce a framework that works for all the variants. Also, many LALMs do not have a decoder that maps back to audio space. Lots just do audio-to-text. Only a few models are fully speech-to-speech (some are trained end-to-end, and others just put a TTS module on end). It is important to talk about these. Furthermore, why can’t you use a straight-through estimator or Gumbel softmax to differentiate through the discretisation instead of the dual optimisation approach? I need more motivation to believe this is necessary.
    - Also, is gradient shattering a well-known term? A quick search gets something different: https://arxiv.org/abs/1702.08591. Perhaps the problem could just be called “Non differentiable audio tokenisation” or similar? I don’t think the dual optimization method is novel, it would be good to find the original paper that implements something like this. PErhaps it would be in the VQVAE literature?
- **Lack of threat model:** I’d like to see your threat model go into depth more about why you focus on white-box attacks and why you need stealthiness constraints. E.g., you can just apply existing white-box attacks to text LLMs already and get bad outputs; why do we care about LALM defense when text isn’t solved? Isn’t an attack that elicits a harmful jailbreak that isn’t “stealthy” also a success from the red team’s perspective? Why does it need to be understandable? These can be addressed in your threat model. Also, you mention in related work that LALMs shouldn’t be deployed widely if they are not robust, but releasing them as closed source is fine since you can’t attack with AdvWave.
- **Presentation of equations, figures, and results needs to be polished:**
    - Figure 1: Phase 1 would be nicer on the left. A brief intuition on what each loss is trying to achieve in the caption would be helpful
    - Section 3.2, in general, is very hard to follow along. L_retent is talked about lot before being explained. Include an intuitive explanation earlier. You introduce the notation for the size mappings of each component, but this makes it more confusing, in my opinion. I would put this in the appendix.
    - Section 3.5 - There is lots of repetition of equations here (e.g. equ 7 is the same as 5 and 6 similar to 1), it would be great if it could be folded into the other sections for conciseness
    - I’m not sure what the perk of having ASR-W is in addition to ASR-L. Often, LLMs are still jailbroken if they say, “I’m sorry,” so I’d expect ASR-W to have many false negatives. It would be good to manually check the false positive rate of ASR-L.
    - Figures 2 & 3 need axes labels and should use a color-blind friendly palette (without gradients). Figure 4 has text that is too small.
- **Related work is majorly lacking citations and doesn’t contrast with AdvWave:**
    - Add related work to white-box attacks on VLMs - your work is very comparable to how people jailbreak VLMs, e.g., https://yunqing-me.github.io/AttackVLM/ , https://arxiv.org/pdf/2306.13213, https://arxiv.org/pdf/2402.02309. Also, vocalising the request is similar to putting typographic text into images (like FigStep, Images are Achilles Heel of Alignment, Jailbreak in pieces)
    - Add related work to white-box attacks on STT models - this is also very relevant, especially the imperceivable constraints. e.g. “Audio adversarial examples: Targeted attacks on speech-to-text”, “There is more than one kind of robustness: Fooling whisper with adversarial examples”.
    - There are many more papers than I provide here, and I’d recommend doing a proper literature review.
    - LALM section - I would cut the section around concerns of misuse. This should be discussed in the intro. You should cite frontier models like Gemini and GPT-4o advanced voice mode.
    - Jailbreak attacks on LLMs section - you should cite https://arxiv.org/abs/2404.02151
- **Adaptive target search seems overly complicated:** why did optimising just for “sure” as the first token not work? This works in VLM literature. When comparing to optimizing for “sure”, did you use a prompt like in https://arxiv.org/abs/2404.02151? If not, optimizing for “sure” alone may be much weaker. I’d expect if you did this, the ASR would increase. Essentially, using an “adaptively search optimisation target,” you find a good starting point, but prompting the model to start the response with “Sure, here is…” might mean you don’t need this component. Also, why can’t you find a target string from another jailbroken LLM even if it has a very different structure to the output of the LALM? Shouldn’t gradient-based approaches still be able to change the model to output this?

**Questions:**

Have you thought about measuring how your attacks transfer between models? I’d love to see transferability in your work since the threat model I think is most concerning is people finding white-box attacks on open-source models that transfer to more powerful closed-source models. See examples here: https://arxiv.org/abs/2403.09766 , https://arxiv.org/abs/2407.15211

Small discussion point on using LALMs. Most of the field uses VLMs for vision language models, so do you think using ALMs would be a better acronym to popularise in the field?

I have weaved most of my questions into the weaknesses section. I think this paper has the potential for a much higher rating (especially given the timeliness of getting attacks working on LALMs, which is a neglected area of the adversarial attack literature), but not in its current form. I am happy to increase my score if the weaknesses I highlighted are addressed.

---

> ### Author Response · Authors · 2024-11-25
> **Response to Reviewer 7mqQ (Part 1)**
>
> We appreciate the reviewer's thoughtful feedback on our paper. Below, we included additional comments to further improve our work.
>
> > Q1: AdvWave is not the first approach to jailbreaking LALMs.
>
> Thank you for the comment! We claim that AdvWave is the first **white-box** attack against LALMs in our revised manuscript.
>
> > Q2: More clarifications on the stealthiness constraint.
>
> Thank you for the insightful question!
>
> [Rationale for Enforcing Stealthiness] The motivation for enforcing stealthiness stems from both empirical observations and insights from jailbreak literature. Without this constraint, optimized adversarial audio—while effective—often exhibits unnatural, screechy qualities. These anomalies draw undue attention from human auditors and increase the likelihood of being flagged or filtered by noise-detection systems. To illustrate this, we include examples of adversarial audio without stealthiness constraints in the supplementary material. By enforcing stealthiness, we aim to produce adversarial audio that sounds natural, reducing suspicion and bypassing noise filters. This approach parallels advancements in text-based jailbreaks, where recent studies [1,2] enhance fluency and readability to evade perplexity-based filters. To address this, we expanded the discussion in Section 3.1 and added relevant work [3], highlighting that enforcing stealthiness is also considered in adversarial attacks against DNN-based audio models.
>
> [Clarifications on Stealthiness Scores of Baselines] We would like to clarify that the baselines (GCG-Trans, BEAST-Trans, and AutoDAN-Trans) optimize adversarial suffixes in text modalities, which are subsequently converted into corresponding audio suffixes using TTS models. As a result, the stealthiness scores of these baselines are not perfect (i.e., not 1.0). In contrast, vanilla generation achieves a perfect stealthiness score of 1.0 because it does not modify the original query. However, adversarial audio suffixes, including those generated by AdvWave, typically reduce stealthiness due to the additional adversarial content introduced.
>
> [Rationale for Considering Adversarial Audio Suffixes] In line with approaches from jailbreak literature [4], where adversarial suffixes are appended rather than modifying original tokens, our reasoning is as follows: altering the original query risks introducing semantic changes, potentially leading to false positive jailbreaks. For instance, changing the query “How to use a gun for fun?” to “How to use a water gun for fun?” may prompt a concrete and innocuous response, which would not constitute a successful jailbreak. Therefore, in the context of LALM jailbreaks, we prioritize preserving the original query to maintain its semantics and minimize the risk of false positives.
>
> [1] Guo, Xingang, et al. "Cold-attack: Jailbreaking llms with stealthiness and controllability." ICML 2024.
>
> [2] Sadasivan, Vinu Sankar, et al. "Fast Adversarial Attacks on Language Models In One GPU Minute." ICML 2024.
>
> [3] Qin, Yao, et al. "Imperceptible, robust, and targeted adversarial examples for automatic speech recognition." International conference on machine learning. PMLR, 2019.
>
> [4] Zou, Andy, et al. "Universal and transferable adversarial attacks on aligned language models." arXiv preprint arXiv:2307.15043 (2023).
>
> > Q3: Lack of relevant adaptive black-box baselines.
>
> Thank you for the valuable feedback! We select the three transfer-based baselines since they all do not need gradient information and show variability in the readability of jailbreak prompts that may affect cross-modality transferability. We further add another strong baseline PAIR attack which leverages a red-teaming LLM to refine the jailbreak suffix with black-box feedback of audio-language models. We compare PAIR and AdvWave in Table A. The results show that AdvWave still outperforms PAIR significantly, highlighting the effectiveness of white-box optimization with AdvWave.
>
> Table A: ASR-W/ASR-L of PAIR attack and AdvWave attack on different audio-langauge models.
>
> | | SpeechGPT | Qwen2-Audio | Llama-Omni |
> | - | - | - | - |
> | PAIR | 0.064 / 0.013 | 0.462 / 0.362 | 0.753 / 0.578 |
> | AdvWave | 0.643 / 0.603 | 0.891 / 0.884 | 0.981 / 0.751 |

---

> ### Author Response · Authors · 2024-11-25
> **Response to Reviewer 7mqQ (Part 2)**
>
> > Q4: Lack of clarity on LALM architecture differences.
>
> Thank you for the thoughtful comment! The term "gradient shattering" originates from [5], where it is defined as nonexistent or incorrect gradients caused either intentionally through non-differentiable operations or unintentionally due to numerical instability. We acknowledge that not all LALMs suffer from gradient shattering. In cases where this issue is absent, adaptive adversarial target search and classifier-guided stealthiness control can be directly applied in an end-to-end optimization framework.
> That said, when gradient shattering presents additional technical challenges for jailbreak, our dual-phase optimization approach serves as a solution. The dual-phase optimization in AdvWave can be viewed as a specific instance of alternating direction methods [6], where different variables are optimized alternately to facilitate optimization.
> Additionally, we clarify that since all LALMs output text, the adversarial loss in AdvWave is applied to the text modality, making our framework broadly applicable to all LALMs. We have incorporated these discussions in Section 3 of the revised manuscript.
>
> [5] Athalye, Anish, Nicholas Carlini, and David Wagner. "Obfuscated gradients give a false sense of security: Circumventing defenses to adversarial examples." International conference on machine learning. PMLR, 2018.
>
> [6] Boyd, Stephen, et al. "Distributed optimization and statistical learning via the alternating direction method of multipliers." Foundations and Trends® in Machine learning 3.1 (2011): 1-122.
>
> > Q5: More clarifications on the threat model.
>
> Thank you for the question! We have added clarifications regarding the importance of enforcing stealthiness in response to Q2. We consider both white-box and black-box optimization to be practically significant. As the widespread deployment of large models continues, it becomes increasingly challenging to centralize these models on a single server. For example, deploying a model on a mobile device effectively creates a white-box scenario for users. Therefore, we argue that studying white-box jailbreaks against LALMs is a meaningful research direction to inform and enhance the future alignment of these models.
> > Q6: Presentation issues.
>
> Thank you for the suggestions! We improved the presentation in our revision based on the comments.
>
> > Q7: Missing related work.
>
> Thank you for the suggestion! We added a literature review on VLM jailbreak and adversarial attack on STT model in Section 2.
>
> > Q8: Adaptive target search seems complicated.
>
> Thank you for the insightful comment. We acknowledge that prior work, such as [1], employs manually designed adversarial optimization targets, which can be labor-intensive and time-consuming. To address this, we propose an adaptive target search algorithm comprising three key components: object detoxification, response collection, and pattern summarization. While the algorithm introduces the overhead of three additional model forward passes, this is minimal compared to the thousands of forward passes required for optimization, making it both efficient and worthwhile. We have incorporated this discussion into the revised manuscript.
>
> [1] Andriushchenko, Maksym, Francesco Croce, and Nicolas Flammarion. "Jailbreaking leading safety-aligned llms with simple adaptive attacks." arXiv preprint arXiv:2404.02151 (2024).
>
> > Q9: Cross-model transferability evaluation.
>
> Thank you for the insightful comment! We evaluate the attack success rates (ASR) of adversarial audio optimized on white-box audio-language models when tested on a Realtime API using the Advbench-Audio dataset. The results, presented in Table B, indicate that the transferability of the optimized audio is limited. This may stem from significant discrepancies in audio encoders, model architectures, and other underlying factors. Developing more transferable attack methods for audio-language models remains an open direction for future research.
>
> Table B: ASR-W and ASR-L of adversarial audio optimized on white-box audio-language models on Realtime API on Advbench-Audio dataset.
>
> | Source model | SpeechGPT | Qwen2-Audio | Llama-Omni |
> | - | - | - | - |
> | ASR-W | 0.205 | 0.157 | 0.057 |
> | ASR-L | 0.046 | 0.024 | 0.023 |
>
>
>
> > Q10: Naming of ALM.
>
> Thank you for the suggestion! We assume ALM is indeed better to understand and memorize. We adopted the naming of ALM in the revision.

---

> > ### Comment · Reviewer_7mqQ · 2024-11-25
> >
> > Thanks for the additional experiments and paper edits.
> >
> > > This unnatural quality draws undue attention from human auditors and risks being flagged or filtered by noise-detection systems.
> >
> > I’m still not convinced by this. If there was a human auditor in the loop then they’d instantly know an adversary is misusing the ALM because they can hear the person asking for harmful information. As for noise detection systems, I’m not aware of any guardrails that exist currently that do this so trying to solve this problem doesn't make sense. If you want this as your motivation, then I think you need to implement your own noise detection system and verify that it works well without false positives on benign data. Personally, I think you should reframe your paper to care about ”misuse” as a threat model where stealthiness is not a constraint your algorithm needs to care about.
> >
> > Thanks for supplying the supplementary material and your method reduces the noiseyness of the audio suffix well. However, I am still not convinced by the motivation.
> >
> > > We would like to clarify that the baselines (GCG-Trans, BEAST-Trans, and AutoDAN-Trans) optimize adversarial suffixes in text modalities, which are subsequently converted into corresponding audio suffixes using TTS models. As a result, the stealthiness scores of these baselines are not perfect (i.e., not 1.0). In contrast, vanilla generation achieves a perfect stealthiness score of 1.0 because it does not modify the original query.
> >
> > This doesn’t make much sense to me as a metric because the baselines (e.g. AutoDAN-Trans) are just as stealthy as the vanilla generation in terms of audio quality. I don’t think you can claim they are less stealthy due to the content of the text being spoken, as the vanilla harmful request is not stealthy at all from a text perspective, either.
> >
> >
> > > We have incorporated [ALM architecture differences] discussions in Section 3 of the revised manuscript.
> >
> > I do not see a clear explanation of how different ALMs differ and which architectures your method works for. This is crucial, in my opinion.

---

> > > ### Author Response · Authors · 2024-11-26
> > > **Follow-up discussion with Reviewer 7mqQ**
> > >
> > > We thank the reviewer’s valuable feedback!
> > >
> > > > Concerns regarding the stealthiness enforcement.
> > >
> > > We acknowledge that most of the jailbreak literature focuses on "misuse" as the primary threat model, where stealthiness is not a significant concern. Only a few follow-up studies have explored enhancing the fluency or readability of jailbreak prompts as secondary benefits to bypass specific guardrails. Consequently, in the revised version of our paper, we will de-emphasize the stealthiness constraint in the threat model and methodology sections. Instead, we will emphasize that the AdvWave framework is flexible and can incorporate additional stealthiness constraints as by-products when specific noise detection mechanisms are in place.
> > >
> > > > ALM architecture discussion.
> > >
> > > Thanks for the comment! We highlight the application of AdvWave on ALMs with different architectures in Section 3.5 in the current version.

---

> > > ### Author Response · Authors · 2024-12-03
> > >
> > > Dear Reviewer 7mqQ,
> > >
> > > As the discussion period draws to a close, we would like to kindly inquire if you have any further questions or feedback regarding our work. Your insights and suggestions have been valuable in helping us improve our work.
> > >
> > > Additionally, we hope you might consider reevaluating your rating in light of the updates and clarifications provided during the rebuttal process.
> > >
> > > Thank you again for your time and thoughtful review.
> > >
> > > Sincerely,
> > > Authors of AdvWave

---

### Official Review · Reviewer_vSXc · 2024-11-03

**Soundness:** 4
**Presentation:** 3
**Contribution:** 3
**Rating:** 8
**Confidence:** 5

**Summary:**

The authors introduce a novel jailbreak framework for optimising audio jailbreaks against audio-language models (ALMs). They overcome challenges in the design of ALMs: namely 1.) they find a dual-phase training process so they can optimise attacks even through discretisation operations in the tokeniser, and 2.) they develop an adaptive search method to find more flexible adversarial targets.

Finally, the authors introduce a realistic constraint on their work: that the audio jailbreaks are stealthy. They operationalise this as having human and ALM-based classifiers independently score the audio input for signs that it was adversarially tampered-with. The authors claim (it's hard without hearing audio samples myself) that their jailbreaks are hence indistinguishable from normal urban noise (e.g. a car horn).

**Strengths:**

This paper tackles a important and as yet unaddressed issue in jailbreak literature, and does so with sensitivity to realism. I am particularly impressed with the authors' operationalisation of stealthiness as urban noise (pending audio samples that I can listen to when the paper is out). The authors' use of human judgment to verify and counterweight their classifier (itself a potentially valuable contribution to audio-jailbreak defense) strengthens my confidence in these results even if I can't aurally verify them myself.

The results of their optimisation are strong. Their ASR results are comparable to or exceed the other audio-jailbreak papers I know of that were submitted to ICLR.

The methods developed to optimise jailbreaks against audio-models, given the challenges the authors list, are valuable and novel contributions themselves. In particular the method for adversarial optimisation target search seems to me to strengthen the jailbreak method over the baselines they test against. For example, GCG is reasonably well-known for optimising for outputs such as "Sure," even if immediately followed with "I can't help with that." The adaptivity and greater detail of the jailbreak targets listed in the appendix seem to me to increase the likelihood that jailbreaks listed as successful in this paper do in fact contain jailbroken information. I'm also given increasing confidence in the evaluations of this paper by the authors' use of both a word-based classifier that detects refusal strings, and an LLM graded response.

**Weaknesses:**

While I'm overall very positive on this paper, I'm a little underwhelmed by the baselines. I would expect that the adversarial perturbations of GCG and BEAST to be quite brittle to being converted to spoken text and then fed into an ALM. These are worthwhile baselines to run, but more semantically-natural baselines like AutoDAN would have pushed the paper even further. The authors acknowledge the difficulty and novelty of introducing audio-based adaptive attacks, like transfers of PAIR or TAP: I would have been very excited to see the authors tackle adaptive jailbreaks in the audio domain, but understand why for reasons of cost and difficulty that this might not be feasible - though I am aware of an unpublished audio implementation of PAIR.

I think Fig 1 is quite challenging to parse. I would rather it be simplified quite a lot more before final release. In particular, I think there is too much text annotating Phase II, even if helpful for diving deeper into the method. I would prefer at least a much more abstracted version of the figure, without reference to variables from Equation 1, and with the annotation retooled to explain how the different branches refer to each other. At the moment I think it's too hard to understand without continuous reference to Equation 1, and the figure struggles to explain itself on its own.

**Questions:**

1. Did you try other audio-transcribed jailbreak classes, including more naturalistic text like in Zheng et al's persuasion paper? [1]
2. What made you think GCG and BEAST were strong baselines when translated into audio?
3. Did you attempt your jailbreaks on any versions of Gemini or 4o? To my understanding some of the more capable models are only trained to recognise speech data - which would presumably make your noise perturbations less effective?
4. Who were the humans judging your stealthiness? was there a more detailed rubric you can share?




[1] https://arxiv.org/abs/2401.06373

---

> ### Author Response · Authors · 2024-11-25
> **Response to Reviewer vSXc**
>
> We appreciate the reviewer's thoughtful feedback on our paper. Below, we included additional comments to further improve our work.
>
> > Q1: Baseline selection is not strong and lack motivations.
>
> Thank you for the valuable feedback!  We select the three transfer-based baselines since they all do not need gradient information and show variability in the readability of jailbreak prompts that may affect cross-modality transferability. We further add another strong baseline PAIR attack which leverages a red-teaming LLM to refine the jailbreak suffix with black-box feedback of audio-language models. We compare PAIR and AdvWave in Table A. The results show that AdvWave still outperforms PAIR significantly, highlighting the effectiveness of white-box optimization with AdvWave.
>
> Table A: ASR-W/ASR-L of PAIR attack and AdvWave attack on different audio-langauge models.
>
> | | SpeechGPT | Qwen2-Audio | Llama-Omni |
> | - | - | - | - |
> | PAIR | 0.064 / 0.013 | 0.462 / 0.362 | 0.753 / 0.578 |
> | AdvWave | 0.643 / 0.603 | 0.891 / 0.884 | 0.981 / 0.751 |
>
> > Q2: Improvement of Figure 1.
>
> Thank you for the comment! We improve the clarity of Figure 1 following the suggestion.
>
> > Q3: Jailbreak attempts on Gemini and GPT-4o.
>
> Thank you for the insightful comment! We evaluate the attack success rates (ASR) of adversarial audio optimized on white-box audio-language models when tested on a Realtime API using the Advbench-Audio dataset. The results, presented in Table B, indicate that the transferability of the optimized audio is limited. This may stem from significant discrepancies in audio encoders, model architectures, and other underlying factors. Developing more transferable attack methods for audio-language models remains an open direction for future research.
>
> Table B: ASR-W and ASR-L of adversarial audio optimized on white-box audio-language models on Realtime API on Advbench-Audio dataset.
>
> | Source model | SpeechGPT | Qwen2-Audio | Llama-Omni |
> | - | - | - | - |
> | ASR-W | 0.205 | 0.157 | 0.057 |
> | ASR-L | 0.046 | 0.024 | 0.023 |
>
>
> > Q4: More details on human judge of stealthiness score.
>
> Thank you for your question. The process for human evaluation of the stealthiness of adversarial audio is designed to assess how imperceptible the adversarial modifications are to a listener. Specifically, a group of domain experts are instructed as follows:
> “You will be presented with two audio clips: the first is the original audio, and the second is its adversarially modified version. Please rate how likely the second audio clip (adversarial audio) introduces only natural background noise as opposed to significant distortions or unnatural artifacts compared to the original audio. Your rating should reflect this likelihood on a scale from 0 to 1, where 0 means 'completely unnatural or obviously manipulated,' and 1 means 'indistinguishable from natural background noise.'”

---

> > ### Comment · Reviewer_vSXc · 2024-11-25
> >
> > > Q1: Baseline selection is not strong and lack motivations.
> >
> > I'm very satisfied now that you're comparing to a much stronger baseline.
> >
> > > Q2: Improvement of Figure 1.
> >
> > The new figure is much easier for me to parse.
> >
> > > Q3: Jailbreak attempts on Gemini and GPT-4o.
> >
> > I understand that since your method is whitebox, it's not possible to do anything other than attempt to transfer attack other models. Do you think that makes this a substantially weaker / less realistic jailbreaking method? How is this change reflected in your threat model for your attack method?
> >
> > > Q4: More details on human judge of stealthiness score.
> >
> > This is helpful context that should be in the paper. I'm not sure how robust the method described is, but nor am I sure of what a better method would be for labelling how stealthy your audio transformations are. Were the authors of the paper some of the judges? How many judges did you have? What did you do to try and make this a repeatable process? What would you have done to make the judging of your audio more consistent/systematic?

---

> > > ### Author Response · Authors · 2024-11-26
> > > **Follow-up discussion with Reviewer vSXc**
> > >
> > > We thank the reviewer’s thoughtful comments again!
> > >
> > > > Practicality of AdvWave as a white-box jailbreak method.
> > >
> > > We consider both white-box and black-box optimization to be practically significant. As the widespread deployment of large models continues, it becomes increasingly challenging to centralize these models on a single server. For example, deploying a model on a mobile device effectively creates a white-box scenario for users. Therefore, we assume that studying white-box jailbreaks against LALMs is a meaningful research direction to inform and enhance the future alignment of these models.
> > >
> > > > Human judge details.
> > >
> > > The details of the human judgment process are provided in Appendix A.5 and referenced in Section 4.1. Currently, two human annotators (the paper's authors) evaluate all audio clips, and the final human judgment score is calculated as the average of their scores. To enhance the annotation process, additional annotators, such as those from platforms like Amazon Mechanical Turk, could be hired for more comprehensive labeling.

---

> > > > ### Comment · Reviewer_vSXc · 2024-11-27
> > > >
> > > > Thanks to the authors for their patient and detailed responses!
> > > >
> > > > > increasingly challenging to centralize these models on a single server
> > > > I agree we can expect e.g. more open source deployment / weight leaks, but your specific example of a deployment on a mobile device seems much more uncertain (which is low stakes for the purposes of this argument).
> > > >
> > > > > additional annotators, such as those from platforms like Amazon Mechanical Turk, could be hired for more comprehensive labeling.
> > > >
> > > > I'd be excited about future, more systematic evaluation - especially for something as interesting and important as the way you operationalise your stealthiness constraint. That you already also use an ALM to detect suspiciousness of your audio samples makes me confident enough in your more minimal use of human judgment.
> > > >
> > > > I think with this discussion my overall grade for the paper remains the same but my confidence has gone up. Updating my review to reflect this.

---

### Official Review · Reviewer_Eqtj · 2024-11-04

**Soundness:** 3
**Presentation:** 3
**Contribution:** 4
**Rating:** 8
**Confidence:** 5

**Summary:**

This paper presents a gradient-based jailbreak attack against Large Audio Language Models (LALM). The proposed method optimizes an adversarial audio suffix that bypasses the safety alignment of the LALM and causes it to produce harmful outputs. To account for the discretization performed to convert continuous audio representations into discrete tokens, a "dual-phase" optimization method is proposed whereby, first, the discrete token sequence is optimized to produce the desired harmful output and then the audio suffix is optimized to yield the discrete audio token sequence. Additionally, an adaptive search procedure is proposed to determine the best target for the adversarial loss optimization, and a loss component is introduced to make the adversarial suffix resemble a given environmental sound. Results show that compared to baselines the proposed approach greatly improves attack success rates.

**Strengths:**

1. The paper is well written and generally clear
1. The proposed approach is novel and fills an important gap in the current literature.
1. The proposed attack is successful on diverse models which indicates its generalizability
1. Using an audio classification loss to make the adversarial suffix resemble natural sounds is an interesting and novel approach

**Weaknesses:**

1. The -Trans baselines seem to weak because these attacks tend to introduce symbols, like special characters, punctuations and emojis, that are not vocalizable so it is expected that generating speech from them will produce weak results. I recommend presenting results for the text-only attack along with the -Trans attack. This way the actual advantage of exploiting the auditory modality will become apparent.
   1. A better baseline could be to adversarially attack an ASR model that uses the same audio encoder as the LALM such that the target transcription is the text-only attack string.

1. More details about the human evaluation score ($S_{\text{Human}}$) are needed, including the number of raters, inter-rater agreement, and did all raters rate all the test audios.
1. The normalization used for the stealth scores seems to be weight the components unfairly. The NSR and cosine are normalized by their theoretic maximum, while the human score is unnormalized so if the actual NSR and cosine scores occupy a smaller range then their contribution to the score will be penalized. A better normalization scheme might be to normalize the mean to 0.5 and standard deviation to 0.125.
1. The presentation can be improved:
   1. Phase II is to the left of Phase I in Figure 1. I suggest reorganizing it to make it appear to the right.
   1. The phrase "gradient shattering" or "shattered gradients" is confusing here because in prior work it refers to the specific phenomenon that as neural networks become deeper their gradients resemble white noise [1]. The particular phenomenon of relevance in this study is generally referred to as "gradient obfuscation" or "obfuscated gradients".
   1. The phrase "retention loss" is confusing because it is not clear what is being retained. The target discrete token sequence can not be "retained" because the encoder currently does not output it and it is being optimized to do so. Perhaps, "alignment loss" or "sequence loss" might be better.
   1. It is not clear from equation 2 that only a suffix is being optimized. It appears that the entire audio is being optimized.


[1] Balduzzi, David, et al. "The shattered gradients problem: If resnets are the answer, then what is the question?." International conference on machine learning. PMLR, 2017.

**Questions:**

1. Why is noise-to-signal ratio used instead of the more common signal-to-noise ratio? Is it computed in a similar manner as SNR? The normalization and subtraction yields a quantity that is proportional to SNR so perhaps its simpler to just use SNR.
1. How exactly is $S_{\text{Mel-Sim}}$ computed? The mel spectrogram is a matrix so how exactly is the cosine similarity computed?
   1. Why is cosine similarity used instead of L2 distance that is commonly used to compare mel spectrograms? I am not sure if the cosine similarity has a reasonable interpretation for mel spectrograms.

**Details Of Ethics Concerns:**

The paper proposes a jailbreak attack against Large Audio Language Models that can enable users to extract harmful information from these models cause them to respond to other users in a harmful manner.

---

> ### Author Response · Authors · 2024-11-25
> **Response to Reviewer Eqtj**
>
> We appreciate the reviewer's thoughtful feedback on our paper. Below, we included additional comments to further improve our work.
>
> > Q1: The Transfer baselines are not strong.
>
> Thank you for the valuable feedback! We select the three transfer-based baselines since they all do not need gradient information and show variability in the readability of jailbreak prompts that may affect cross-modality transferability. We further add another strong baseline PAIR attack [1] which leverages a red-teaming LLM to refine the jailbreak suffix with black-box feedback of audio-language models. We compare PAIR and AdvWave in Table A. The results show that AdvWave still outperforms PAIR significantly, highlighting the effectiveness of white-box optimization with AdvWave.
>
> Table A: ASR-W/ASR-L of PAIR attack and AdvWave attack on different audio-langauge models.
>
> | | SpeechGPT | Qwen2-Audio | Llama-Omni |
> | - | - | - | - |
> | PAIR | 0.064 / 0.013 | 0.462 / 0.362 | 0.753 / 0.578 |
> | AdvWave | 0.643 / 0.603 | 0.891 / 0.884 | 0.981 / 0.751 |
>
> [1] Chao, Patrick, et al. "Jailbreaking black box large language models in twenty queries." arXiv preprint arXiv:2310.08419 (2023).
>
> > Q2: More details on the human study and stealthiness score computation.
>
> Thank you for your question. The process for human evaluation of the stealthiness of adversarial audio is designed to assess how imperceptible the adversarial modifications are to a listener. Specifically, three domain experts are instructed as follows: “You will be presented with two audio clips: the first is the original audio, and the second is its adversarially modified version. Please rate how likely the second audio clip (adversarial audio) introduces only natural background noise as opposed to significant distortions or unnatural artifacts compared to the original audio. Your rating should reflect this likelihood on a scale from 0 to 1, where 0 means 'completely unnatural or obviously manipulated,' and 1 means 'indistinguishable from natural background noise.'” Therefore, the human evaluation scores are also bounded, so that the score combinations with similarity scores is also reasonable. We will include more details in the final manuscript.
>
> Thank you for the suggestion! We will adopt the concept signal-to-noise ratio in our revision.
> For the spectrogram similarity metric, we expand the mel spectrogram matrix to a vector and then compute cosine similarity. With this metric, we expect that it would be intensity invariant since the intensity stealthiness is already reflected in signal-to-noise ratio scores. We aim to evaluate the shape similarity of waveforms so that cosine similarly would be a better choice.
>
>
> > Q3: Presentation issues.
>
> Thank you for the thoughtful comment! The term "gradient shattering" originates from [5], where it is defined as nonexistent or incorrect gradients caused either intentionally through non-differentiable operations or unintentionally due to numerical instability.
> Moreover, in the revised manuscript, we rename "retention loss" as "alignment loss" and refine Equation 2 to emphasize that we only refine a suffix.
>
> [5] Athalye, Anish, Nicholas Carlini, and David Wagner. "Obfuscated gradients give a false sense of security: Circumventing defenses to adversarial examples." International conference on machine learning. PMLR, 2018.

---

> > ### Comment · Reviewer_Eqtj · 2024-11-26
> >
> > Thanks for your response. I still have the following concerns:
> >
> > 1. In Table A, is PAIR used in the transfer setting, i.e. the text suffix it generates is converted into speech via TTS? If yes, then my concerns about the weak baseline remains. I suggested that in addition to the transfer setting you must show the results in the text-only setting as well to demonstrate the advantage of using the speech-based attack.
> > 1. I am confused by you statement that you want to evaluate the "shape similarity of the *waveforms*" by computing the cosine similarity of the mel-spectrum. Please clarify. Also, is there past works that have used cosine similarity to compare mel-spectrograms? If so, please provide citations.
> >
> > Thank you for providing clarification on gradient shattering. I would recommend adding a sentence to define gradient shattering in the paper in order to avoid any confusion.

---

> > > ### Author Response · Authors · 2024-11-26
> > > **Follow-up discussion with Reviewer Eqtj**
> > >
> > > We thank the reviewer for the additional valuable comments!
> > >
> > > > Clarification of PAIR attack in Table A.
> > >
> > > The PAIR attack represents an adaptive black-box attack against ALMs. Specifically, we employ the PAIR framework, leveraging feedback directly from ALMs. In this process, the red-teaming LLM iteratively refines jailbreak prompts based on feedback from the black-box ALMs. While a TTS model is used to convert text into audio, the feedback originates directly from the target ALMs, ensuring that this approach serves as a strong adaptive baseline.
> > >
> > > > Clarifications on the mel-spectrum Cosine similarity.
> > >
> > > Our choice of Cosine similarity over L2-distance emphasizes the similarity in the shape of the waveform while disregarding variations in magnitude or phase. This approach aligns with the principles applied in the literature on [speaker identification](https://www.iosrjournals.org/iosr-jce/papers/Vol26-issue1/Ser-1/C2601011926.pdf).

---

> > > > ### Comment · Reviewer_Eqtj · 2024-11-27
> > > >
> > > > Regarding the PAIR attack and the transfer baselines. I have to reiterate that I suggested showing the success rate of *text-only* attacks as an additional baseline. From the authors' description, it seems that they have used PAIR in the transfer setup.

---

> > > > > ### Author Response · Authors · 2024-12-02
> > > > >
> > > > > Thank you for the suggestion! We have added the results of applying PAIR exclusively on text modalities, using the jailbreak text as input to the audio-language models. These results, presented in Table A, demonstrate that AdvWave consistently outperforms text-only attacks. This is attributed to its ability to manipulate the audio space, which offers greater complexity compared to the discrete token space. Additionally, we will include text-only attack results for other baselines (GCG, BEAST, AutoDAN) in the final version of the paper.
> > > > >
> > > > > Table A: ASR-W/ASR-L of PAIR (Audio), PAIR (Text), and AdvWave attack on different audio-langauge models.
> > > > >
> > > > > | | SpeechGPT | Qwen2-Audio | Llama-Omni |
> > > > > | - | - | - | - |
> > > > > | PAIR (Audio) | 0.064 / 0.013 | 0.462 / 0.362 | 0.753 / 0.578 |
> > > > > | PAIR (Text) | 0.152 / 0.163 | 0.632 / 0.602 | 0.796 / 0.683 |
> > > > > | AdvWave | 0.643 / 0.603 | 0.891 / 0.884 | 0.981 / 0.751 |

---

> > > > > > ### Comment · Reviewer_Eqtj · 2024-12-03
> > > > > >
> > > > > > Thank you for providing this result. My concerns have been addressed and I have increased my score.
> > > > > >
> > > > > > Good work!

---

### Author Response · Authors · 2024-11-25
**Revision Summary**

We thank all reviewers for their valuable comments and feedback! We are glad that the reviewers found our work novel (the first white-box jailbreak attempt against LALMs) and sound with solid evaluation results. Based on the reviews, we have conducted additional experiments and made the following updates:

1. Expanded discussions on the importance of stealthiness, added supplementary examples, and clarified stealthiness evaluation methods.

2. Introduced the PAIR attack as a strong black-box baseline to emphasize AdvWave’s superior performance.

3. Enhanced explanations on gradient shattering and its implications for AdvWave optimization.

4. Clarified the rationale for appending adversarial suffixes instead of modifying original queries.

5. Improved presentation throughout the manuscript, including figures and equations.

6. Included additional related work on adversarial attacks for speech models and VLM jailbreaks.

7. Expanded discussions on the threat model, highlighting the relevance of white-box jailbreak scenarios.

8. Evaluated cross-model transferability of adversarial audio on Realtime API on the Advbench-Audio dataset.

9. Acknowledged the limitations in adversarial audio transferability and outlined future research directions.

---

### Meta-Review · Area_Chair_T3yh · 2024-12-24

**Metareview:**

This paper proposes a jailbreak attack against Large Audio Language Models (LALM). The method uses a dual-phase optimization method that addresses the non-differentiable issue, enabling effective end-to-end gradient-based optimization. The method also enforces stealthiness constraints on adversarial audio waveforms to avoid being filtered by noise-detection systems. All the reviewers acknowledge the novelty of this work. Some reviewers question the need to maintain stealthy. In my opinion, the authors did a good job answering most of the concerns raised by the reviewers. Therefore, I recommend accept for this paper.

**Additional Comments On Reviewer Discussion:**

Some reviewers question the threat model, i.e., why do we care about whether the audio is stealthy or not? The authors explained that without the constraints, the unnatural audio quality may be flagged or filtered by noise-detection systems. I am convinced by this explanation, and thus, I lowered the weights of the two reviewers who raised this concern and gave low scores.

---

### Decision · Program_Chairs · 2025-01-22

Accept (Poster)